# Voice Interaction Recognition Design in Real-Life Scenario Mobile Robot Applications

**Shih-An Li [1], Yu-Ying Liu [1], Yun-Chien Chen [1], Hsuan-Ming Feng [2,*], Pi-Kang Shen [1] and Yu-Che Wu [1]**

1   Department of Electrical and Computer Engineering, Tamkang University, Taipei City 10650, Taiwan
2   Department of Computer Science and Information Engineering, National Quemoy University, Kinmen County 89250, Taiwan
*   Correspondence: hmfenghmfeng@gmail.com

**Abstract:** This paper designed a voice interactive robot system that can conveniently execute assigned service tasks in real-life scenarios. It is equipped without a microphone where users can control the robot with spoken commands; the voice commands are then recognized by a well-trained deep neural network model of automatic speech recognition (ASR), which enables the robot to execute and complete the command based on the navigation of a real-time simultaneous localization and mapping (SLAM) algorithm. The voice interaction recognition model is divided into two parts: (1) speaker separation and (2) ASR. The speaker separation is applied by a deep-learning system consisting of eight convolution layers, one LSTM layer, and two fully connected (FC) layers to separate the speaker's voice. This model recognizes the speaker's voice as a referrer that separates and holds the required voiceprint and removes noises from other people's voiceprints. Its automatic speech recognition uses the novel sandwich-type conformer model with a stack of three layers, and combines convolution and self-attention to capture short-term and long-term interactions. Specifically, it contains a multi-head self-attention module to directly convert the voice data into text for command realization. The RGB-D vision-based camera uses a real-time appearance-based mapping algorithm to create the environment map and replace the localization with a visional odometer to allow the robot to navigate itself. Finally, the proposed ASR model was tested to check if the desired results will be obtained. Performance analysis was applied to determine the robot's environment isolation and voice recognition abilities. The results showed that the practical robot system successfully completed the interactive service tasks in a real environment. This experiment demonstrates the outstanding performance with other ASR methods and voice control mobile robot systems. It also verified that the designed voice interaction recognition system enables the mobile robot to execute tasks in real-time, showing that it is a convenient way to complete the assigned service applications.

**Keywords:** automatic speech recognition; voice interaction; mobile robot; simultaneous localization and mapping (slam); deep learning

## 1. Introduction

In recent years, artificial intelligence (AI) has led to breakthroughs in linear predictive coding (LPC) and dynamic time warp (DTW) in the field of speech.

The most important breakthrough in the history of speech technologies dealing with large amounts of data is the application of hidden Markov models (HMM). Today's automatic speech recognition models not only have the framework of HMM, but other techniques related to artificial intelligence such as deep neural networks (DNN) or recurrent neural networks (RNN) are also heavily used in this field. The current popular technologies in speech-related fields are broadly classified as automatic speech recognition (ASR), natural language processing, text-to-speech (TTS), speaker verification, and speaker recognition. These can be combined to build more complex applications such as speech-to-speech translation. Voice technology is currently being used in many areas; it is utilized as voice assistants, in ticketing systems, chatting robots, real-time voice translation, etc.

The topic of human–computer interaction has always been of great interest to everyone in the applications of the mobile robot [1], and speech-related technologies are also heavily used in this topic. Human–computer interaction is the interaction and communication between humans and machines. The most intuitive and simplest communication of humans is through speech. In the past, humans have relied on interfaces to operate robots, and it takes time for the elderly or children to learn, and become familiar with how to operate them. If the robot can directly listen and follow the user's voice command to perform service tasks, it will be a lot easier and more convenient for humans. For a more complete description in the advantages of human–robot interaction, please refer to [2].

This research withdrew from the traditional manual interface, but focused on directly controlling the robot through a voice command to perform specific tasks. A human–interactive robot is an inevitable trend. A service robot that has functions of automatic speech recognition (ASR), natural language understanding, image recognition, and interactive communication will approach future development.

The simplest case of the speaker separation model is to disentangle two mixed voice signals with the same dimension of input and output. In recent years, deep clustering [3], TasNet [4,5], and permutation invariant training (PIT) [6] models have become popular speaker separation models. Below is a brief discussion of these models.

Deep clustering uses the concept of k-means clustering, which first converts a sound signal into a spectrum and then, translates the spectrum into a vector. The vector is composed of the units calculated in the spectrum and the surrounding units, so the spectrum is transformed from a flat matrix at the beginning to a matrix-like a cube; thus, one dimension is added. Next, the transformed matrix is divided into two categories using k-means clustering, where the number of speakers is assumed to be two. The k-means clustering method is based on the principle of aggregating similar vectors and the result of clustering produces two sets of masks, which will be multiplied by the original input signal to produce two separate voice signals.

TasNet or the time-domain audio-separation network has two main frameworks: Bi-LSTM and CNN. The original one, which uses Bi-LSTM as the core has a long training time because of the LSTM; thus, CNN was later added to improve the original architecture. TasNet is divided into three main parts: encoder, separator, and decoder. Initially, the sound signal is generated as a matrix by the encoder, which is like a Fourier transform that converts the sound signal into a spectrum; the difference is that the three parts of TasNet are all trained. The encoder output matrix is generated by the separator mask, which is similar to the speaker separation classical framework. Finally, the mask is multiplied with the encoder output and input into the decoder to obtain the separated sound signal.

The idea of permutation invariant training is to first give the speaker separation model and calculate different losses through different output permutations and then, continue training the model after deducing a correct permutation from the losses. Therefore, a separation model is first initialized randomly to obtain a permutation; then, the model is updated and iterated until convergence. This method still needs to be executed with a known number of talkers, but it can already be used for end-to-end purposes.

Automatic speech recognition (ASR) is used to capture the acoustic signal of speech and determine the spoken words by a pattern-matching method. ASR is efficiently and accurately to convert voice signal content into its corresponding text. Its function is a little different from the above-mentioned speaker separation models since they deal with the speaker's vocal signal, while the ASR analyzes the vocabulary content of the voice signal. The natural communication between machine and human can be enhanced by ASR.

The acoustic model (AM) is commonly thought of as the modeling of sound generation. In general, the AM is used as the basic unit for automatic speech recognition, converting speech signals into acoustic states. More precisely, it is the probability of converting a certain acoustic symbol. AM mostly uses the HMM [7], which has a good mathematical structure and is the theoretical basis for the formation of a large number of applications. The HMM is the internal state of this Markov model, while the external world only sees

the output values at each moment, which are the acoustic symbols (acoustic features) for automatic speech recognition.

The pronunciation model contains all the sets of words that need to be processed and their pronunciations; thus, it is also known as the pronunciation dictionary. Normally, the Viterbi algorithm is an efficient approach to decode the acoustic feature probabilities output by the HMM and outputs them as appropriate word strings with a maximum probability.

The purpose of the language model is to combine the output of the pronunciation model into reasonable sentences according to the logic and probability of language usage. It converts a string of letters into a sentence that people can understand.

Traditional automatic speech recognition, as mentioned above, is trained separately using different training data, tuning parameters, optimization strategies, and loss of functions that are not directly transferable, which can be improved with machine learning.

In recent years, the development of machine learning has boomed significantly, which has led to its application in the field of speech. With research, the disadvantages of traditional automatic speech recognition model training have been improved and the end-to-end model and sequence-to-sequence model were found to be reasonable and effective solutions. For example, in 2012, Google developed the RNN-Architecture [8], which incorporates the memory feature of the recurrent neural network (RNN) into the ASR model. Not only did it eliminate the need to align audio and text, but it also achieved an end-to-end effect, jointly optimizing the parameters of the model and allowing the selection of the best phrase for the language model. In 2017, Google added the well-developed attention memory model to the ASR model and developed the LAS model, which can input a string of data and output the data together. Since its development, it has become popular in the field of speech [9]. The sequence-to-sequence deep-learning models are the leading architecture to generate voice recognition applications. Specifically, multiple layers of self-attention blocks are a powerful tool to encode acoustic features [10–13]. The encoded features are then decoded into word sequences using recurrent transducers [14] or attention-based models [8] for the ASR applications. This paper selected a novel sandwich-type conformer model to approach the generic function of the ASR system to control the mobile robot.

Simultaneous localization and mapping (SLAM) has been an important research topic in the field of robot navigation in the last decade. It offers map generation and self-localization abilities to greatly improve the working efficiency of robots in an unknown environment and to avoid collision problems.

Nowadays, many types of sensors are proposed to build maps. The most popular laser sensors, such as the two-dimension laser Hector SLAM [15], combine various algorithms to approach the target of SLAM. This paper used a camera-type sensor in the SLAM application called the vision-based simultaneous localization and mapping (vSLAM) [16,17]. The vSLAM has the advantage of a lower hardware cost and bountiful information. The monocular-type camera only catches a two-dimensional picture and cannot realize the distance of an object.

Due to the fixed axle spacing between two camera lenses, a binocular-type camera uses two camera lenses to capture two pictures and fuses them to calculate the depth of the object. However, it is difficult to perfectly position the camera lens and it consumes high computation resources.

The RGBD-type camera, which uses three color channels (RGB) to capture bountiful information, and the infrared structured light or time of fly (TOF) are used to directly obtain the three-dimensional information (including depth). The efficiency of the RGBD-type camera is very similar to the laser radar.

This paper applied the robot operating system (ROS) platform to develop the robot system. ROS features a distributed frame to generate multiple communications in a topic, service, etc. ROS is compatible with many programs and languages, such as C++, Python, and JavaScript, with a topic or service module to communicate and develop applications.

Another advantage of ROS is that it offers a higher integration and systematically builds individual module examination and verification.

This paper designed an interactive human–robot system with a voice command to control a mobile robot and enable it to complete service tasks. The deep-learning neural network structure was employed for analysis and to allow the mobile robot equipped with an ASR to make an appropriate decision. The mobile robot monitors the current state of the task to approach the required action. A real-time appearance-based mapping (RTAB-MAP) algorithm [17] with a visual odometer was used to construct the three-dimensional maps and allow the robot to gradually move into the desired position in the service environment.

This paper is organized in five parts, where Section 2 explains the human–voice interface system to show the outstanding nature of the proposed methods, Section 3 presents environment map generation through VSLAM, Section 4 completes voice interactive robot control in a real-robot experiment, and Section 5 presents the conclusions.

## 2. Human–Voice Interface System

The audio signal is input through the microphone; the input audio, which can be from multiple people or other noise in the background, is then pre-processed at this stage (called noisy audio from hereon). Next, the speaker separation system is used to separate the audio part of a specific speaker from the noisy audio. The separated speaker-specific audio signals are transformed into speaker command strings through a speech recognition system to provide the robot with decision-making ability.

### 2.1. Speech Pre-Processing

Even if a microphone with a specified pattern is used, the sound signal may still be mixed with background sounds and noises; however, voice recognition only requires segments of human speech. If there is noise or background sound, the accuracy of the recognition will be greatly affected. To remove background sounds, noise, and silent signals for the subsequent ASR, this pre-processing step used voice activity detection (VAD)/speech activity detection (SAD). VAD is mainly used to detect the presence of speech sounds in the current incoming voice signal, and its classical design consists of three steps: first, the sound signal goes through a noise reduction process, such as spectral subtraction [18]; then, it undergoes feature extraction and classifier, which are the focus of VAD.

In this paper, we used the VAD algorithm in WebRTC, which is a multi-feature integrated evaluation that employs the Gaussian mixture model (GMM) [19]. The GMM solves the clustering problem of the samples by estimating the probability density distribution of the sample; thus, it is suitable for VAD:

$$f(x) = \frac{1}{\sigma\sqrt{2\pi}}e^{-\frac{(x-\mu)^2}{2\sigma^2}} \tag{1}$$

$$f(x_k|Z, r_k) = \frac{1}{\sqrt{2\pi}}e^{-\frac{(x_k-\mu_z)^2}{2\sigma^2}} \tag{2}$$

Gaussian distribution is also known as a normal distribution. If the random variable X serves a Gaussian distribution with a location denominator $\mu$ and a scale denominator $\sigma$ then its probability density function can be expressed as Equation (1). The Gaussian distribution in WebRTC is expressed as Equation (2), in which $x_k$ is the selected feature vector that is the energy feature of the six sub-bands just mentioned; and $r_k$ is the combination of the mean $\mu_z$ and variance $\sigma$ parameters, which determine the probability of the Gaussian distribution. $Z = 1$ calculates the probability of speech, $Z = 0$ calculates the probability of noise, and the final classifier uses the likelihood ratio test. If the likelihood ratio is greater than the threshold value, it is considered as speech, which is then output to the speaker separation system.

### 2.2. Speaker Separation System

This paper utilized the speaker separation system [20] to achieve speaker-specific voice separation. The speaker separation system architecture shown in Figure 1 is divided into two main parts: (1) a speaker encoder, which is responsible for extracting the voice pattern characteristics of a specific speaker set by the user; and (2) a voice filter model, which is responsible for retaining and outputting the voice of a specific speaker in a noisy or multi-person environment. The two are trained separately; the details are as follows:

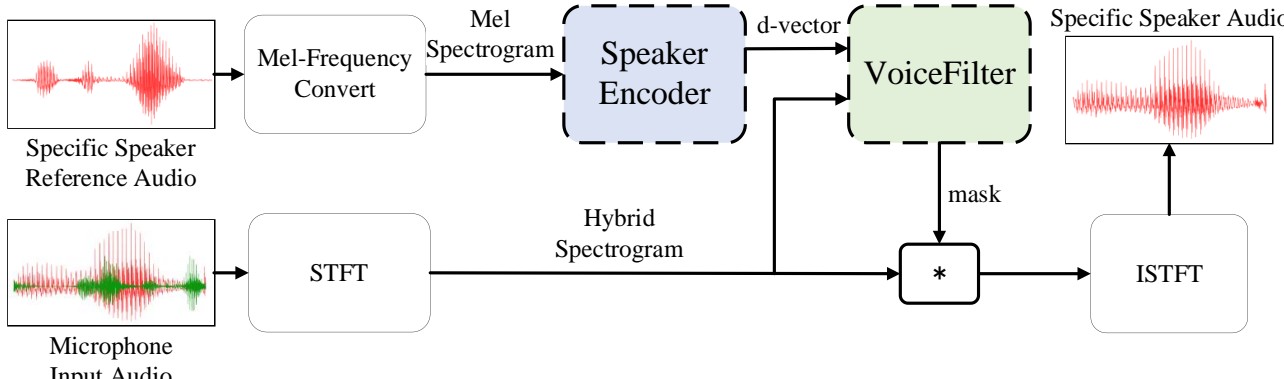

**Figure 1.** Speaker separation system architecture diagram.

(1)     Speaker encoder

The Speaker encoder extracts the embedding that represents the features of a particular speaker's voice pattern, which is called the d-vector. The speaker encoder architecture is a three-layer long short-term memory (LSTM) with generalized end-to-End loss (GE2E) [21]. The input is the Mel-inverted spectrum (MFC) [22] with reference audio conversion, having one frame every 1.6 s with each frameshift at 50%, which is equivalent to 0.8 s. The output is a 256-dimensional d-vector. The speaker encoder architecture is shown in Figure 2.

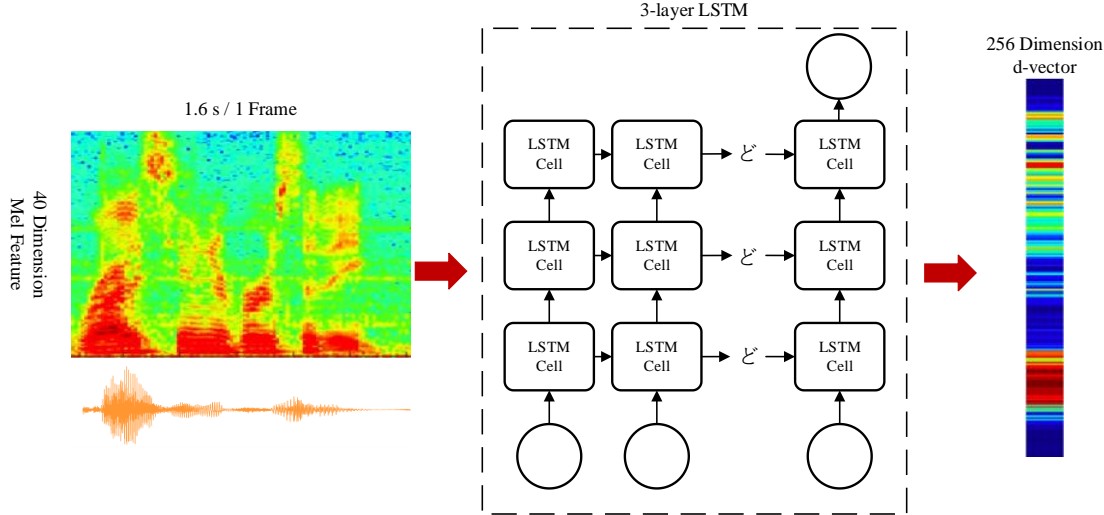

**Figure 2.** Speaker-Encoder Architecture Diagram.

(2)     Voice Filter model

This voice filter part is the main architecture of the system containing 8 layers of convolution layer, 1 layer of LSTM, and 2 layers of FC layer. The training and running process is shown in Figure 3. The input is a mix of audio and the d-vector. Before the mix enters the voice filter, it goes through short-time Fourier transform (SFTF), is converted

into a spectrum map, and is input to the convolution layer. Meanwhile, the d-vector is used as an input for long-term memory (LSTM). There are two reasons for placing the d-vector in the long- and short-term memory (LSTM), but not in the convolutional layer as input. First, the d-vector already has good robustness to represent the vocal features of a particular speaker and does not need to be processed in the convolutional layer; and second, the convolutional layer assumes that time and frequency are aligned. Therefore, two completely different signals cannot be put together as inputs.

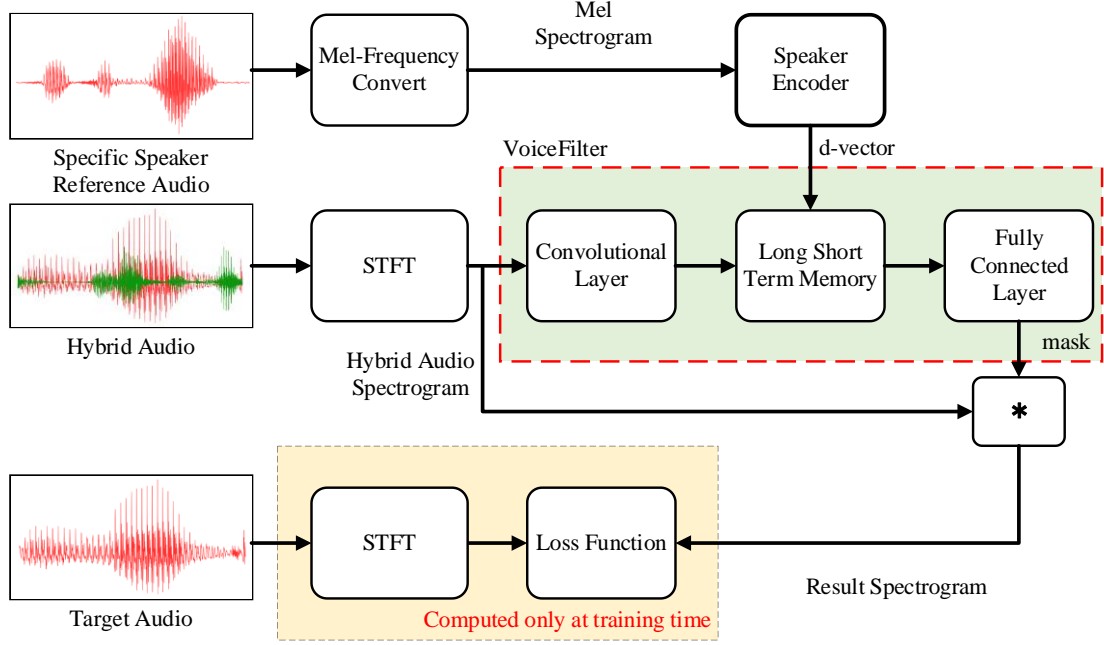

**Figure 3.** Voice filter architecture diagram.

As shown in Table 1, except for the last FC layer (FC) where the activation function is sigmoid, all the other layers are commonly used ReLU. Additionally presented in Table 1 are the parameters of each layer; the loss function uses mean square error (MSE).

**Table 1.** Parameters of voice filter network.

| Layer | Kernel Size | | Dilation | | Filters/Nodes |
|---|---|---|---|---|---|
| | Time | Freq | Time | Freq | |
| Convolution Layer 1 | 1 | 7 | 1 | 1 | 64 |
| Convolution Layer 2 | 7 | 1 | 1 | 1 | 64 |
| Convolution Layer 3 | 5 | 5 | 1 | 1 | 64 |
| Convolution Layer 4 | 5 | 5 | 2 | 1 | 64 |
| Convolution Layer 5 | 5 | 5 | 4 | 1 | 64 |
| Convolution Layer 6 | 5 | 5 | 8 | 1 | 64 |
| Convolution Layer 7 | 5 | 5 | 16 | 1 | 64 |
| Convolution Layer 8 | 1 | 1 | 1 | 1 | 8 |
| LSTM | | | | | 400 |
| FC 1 | | | | | 600 |
| FC 2 | | | | | 600 |

Table 1 shows the design of dilated convolutional layers to extract low-level acoustic features more efficiently. The time and frequency of the spectrum map in speech recognition were the same as the height and width of the image in image recognition.

(3)     Evaluation and Training in Voice Filter model

The Libri speech dataset [23] was used in our voice filter model and its training flow is shown in Figure 4. First, before the model started training, all the parameters and training data of the model were set up in advance; the training data include the training set and test set. Training of the model began after setting one of the parameter settings to "step", which determines the time when the model weights are saved. After the training reached the number of times the step was set, the model weights were saved and immediately substituted into the model for evaluation of the goodness of fit.

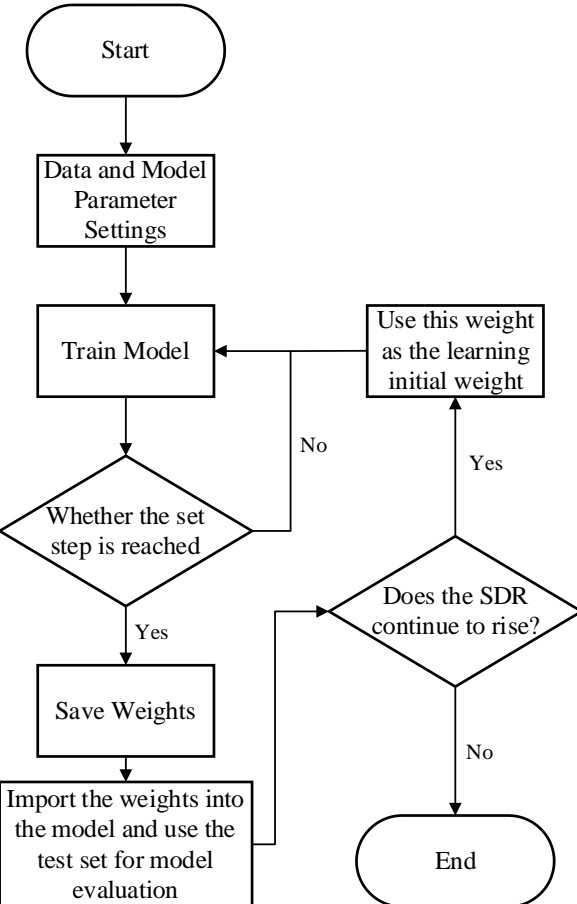

**Figure 4.** Voice filter model train flow chart.

In this paper, the signal-to-distortion Ratio (SDR) is used to evaluate the voice filter model. SDR is commonly used to evaluate signal separation systems [24] and is calculated as the sound intensity ratio in decibels (dB), which determines the error between the projection of the resulting audio and the energy of the clean audio. As shown in Figure 5, the more parallel the two sound vectors are (the more similar the sound is), the higher the signal distortion rate is; unlike the signal-to-noise ratio (SNR), which is not affected by the sound size.

As described in Figure 5 and Table 2, the output audio vector $X^*$ is projected to the vertical direction of the target audio vector $\hat{X}$ to obtain $X_T$, which is parallel to the target audio vector $\hat{X}$. $X_T$ is determined using Equation (3). Finally, Equation (4) is calculated. If the output audio vector $X^*$ is parallel to the target audio vector $\hat{X}$, the larger the SDR is, which means that the sounds are more similar; if the output audio vector $X^*$ is perpendicular to the target audio vector $\hat{X}$, the smaller the SDR is, which means that the sounds are different.

$$X_T = \frac{X^* \cdot \hat{X}}{||\hat{X}||^2} \hat{X} \tag{3}$$

$$SDR = 10 \log_{10} \frac{||X_T||^2}{||X_E||^2} \tag{4}$$

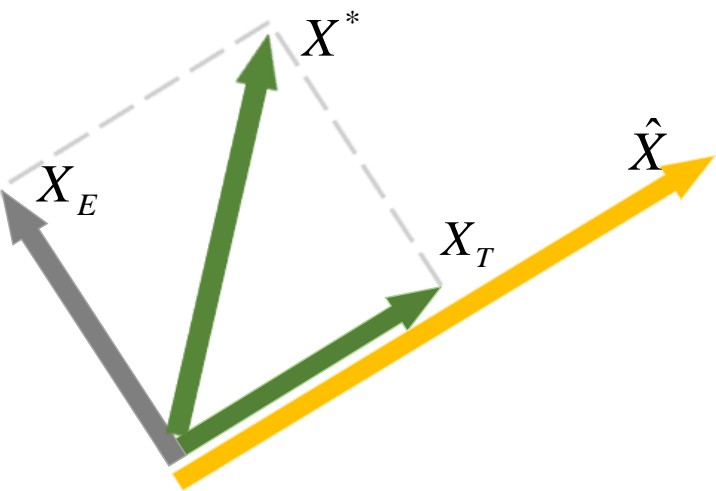

**Figure 5.** SDR vector diagram.

**Table 2.** Meaning of SDR symbol.

| Symbol | Meaning |
|---|---|
| $X^*$ | Vector of output audio |
| $\hat{X}$ | Vector of target audio |
| $X_T$ | $X^*$ project $\hat{X}$ |
| $X_E$ | $X^* - \hat{X}$ |

In this paper, we used the SDR for model evaluation and employed the test set for SDR calculation. When the SDR after model evaluation did not continue to rise after several sets of steps, the model was stopped; conversely, when the SDR continued to rise, we retrained the model with that weight as the initial weight for model learning.

### 2.3. Automatic Speech Recognition (ASR)

In this paper, ASR used the conformer [11] model, which utilized the transformer that is specialized in capturing a large range of feature interaction information, and the CNN specialized in extracting local subtle features; both have contributed to the field of speech and other areas of machine learning in recent years. The conformer model combines the advantages of both for speech recognition.

### 2.3.1. Conformer Model Architecture

The conformer model is a modification of the encoder in the transformer framework, as it incorporates the conformer blocks of the CNN framework. The conformer architecture is shown in Figure 6. The left side is the classic transformer model architecture and the right side (red box) is the internal architecture after the modification. At the start, the spectrum of the audio signal output from the speaker separation system is inputted into the encoder; then, after spectrogram augmentation, it enters linearly in the convolution subsampling, dropout, and the main conformer blocks. Conformer blocks are executed N times according to the input size, just like the encoder in the transformer architecture.

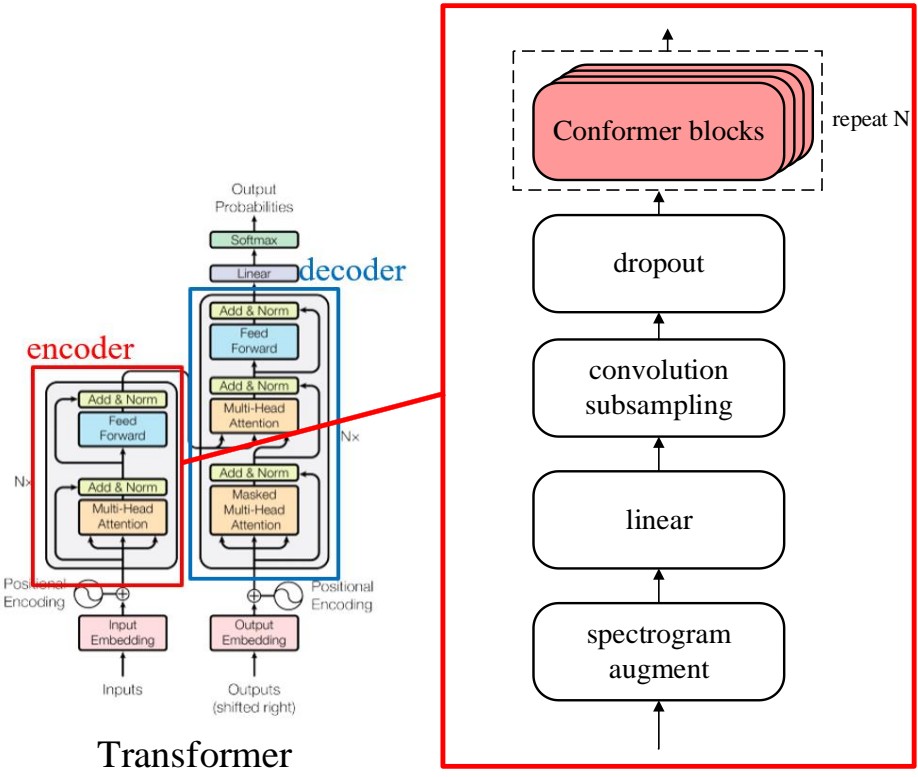

**Figure 6.** Conformer model structure.

The internal structure of the conformer blocks is shown in Figure 7. It consists of three modules: feed-forward module, multi-head self-attention module, and convolution module, each of which uses a residual connection. The structure of the conformer blocks is like a sandwich, where the feed-forward module is located between the multi-head self-attention module and the convolution module, connecting them with each other; however, each feed-forward module only contributes half of its weight. Finally, it enters the post-layer norm for data processing, that is, the post-norm residual unit is utilized. Figure 8 [25] presents the Macaron-net [26] structure, which is formed through Equations (5)–(9). Conformer blocks are represented by x_i for the ith input, y_i for the output, FFN denotes the feed-forward module, MHSA denotes the multi-head self-attention module, and the convolution module is symbolized by Convol.

$$\widetilde{x}_i = x_i + \frac{1}{2}FFN(x_i) \tag{5}$$

$$x_i' = \widetilde{x}_i + MHSA(\widetilde{x}_i) \tag{6}$$

$$x_i'' = x_i' + Conv(x_i') \tag{7}$$

$$\widetilde{x_i''} = x_i'' + \frac{1}{2}FFN(x_i'') \tag{8}$$

$$y_i = layernorm(\widetilde{x_i''}) \tag{9}$$

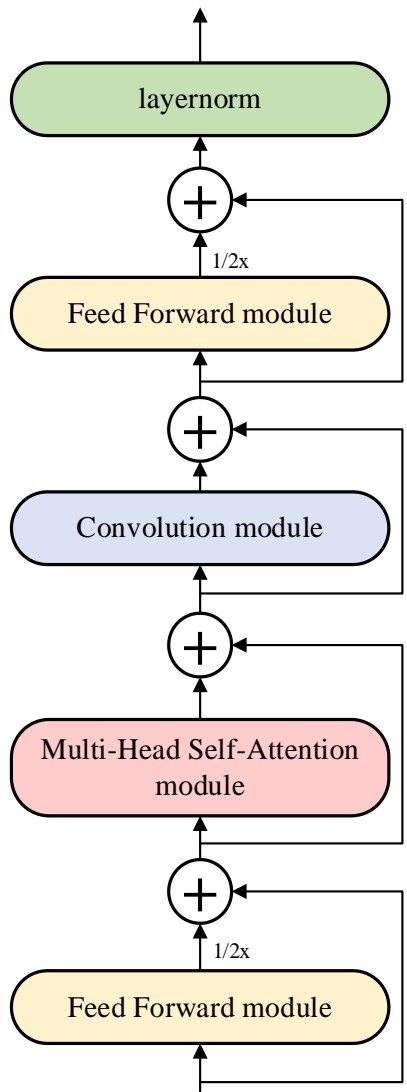

**Figure 7.** Conformer blocks.

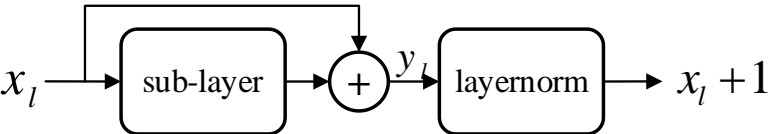

**Figure 8.** Post-norm residual unit.

Multi-head self-attention is the key structure in the transformer, and is a variation of self-attention. It is similar to an RNN, but it can input and output a whole set of data at the same time. Figure 9 shows the structure of the multi-head self-attention module, which uses the relative positional embedding and incorporates the pre-norm residual units with dropout (see Figure 10). Consequently, the convolution and feed-forward modules are also designed using the same structure. The transformer is unable to learn information on the sequence of the positions, so the developers of the transformer proposed to use position embedding for the multi-head self-attention to gain position information. Relative position coding provides better generalization over different input lengths and better robustness to variable speech lengths.

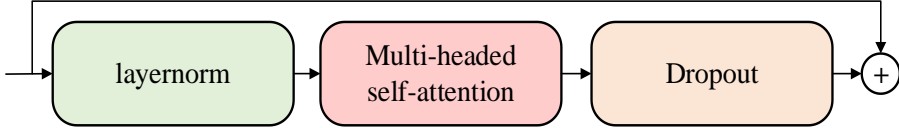

**Figure 9.** Multi-head self-attention module.

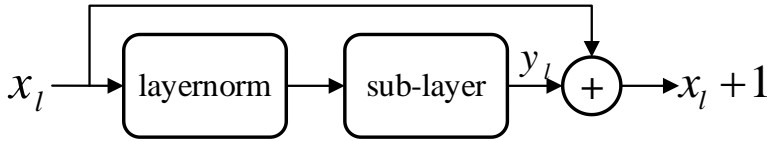

**Figure 10.** Pre-norm residual unit.

### 2.3.2. Convolution Module Design

The convolution module starts from the rating mechanism [27], which consists of a pointwise convolution layer and a gated linear unit (GLU), followed by a 1D depth-wise convolution layer. The batch-norm is arranged after the convolution layer to facilitate model training, and the pre-norm residual unit is used as in the multi-head self-attention module. The convolutional module architecture is shown in Figure 11.

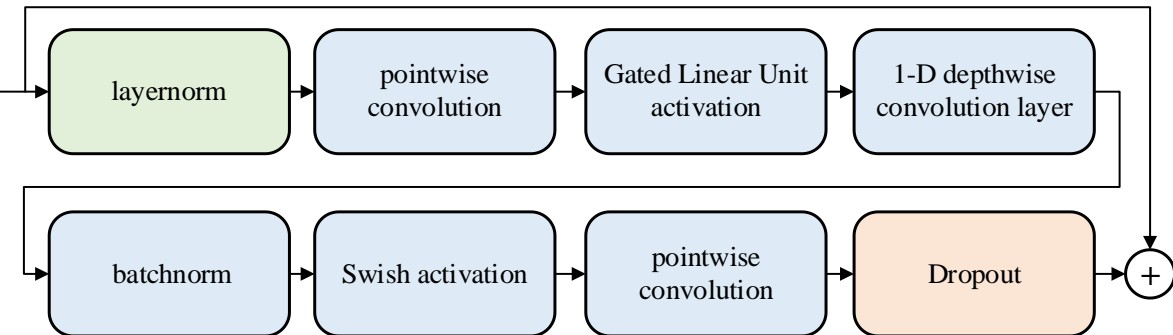

**Figure 11.** Convolution module.

As mentioned, the used depth-wise separable convolution [28] is a new computational structure for CNN developed by Google. It aims to reduce the computational effort of CNN by splitting the original convolutional computation into two parts: pointwise convolutions and depth-wise convolutions, each of which is performed without affecting the output structure to reduce the computation.

### 2.3.3. Feed-Forward Module

As proposed in reference [29], the feed-forward module of the transformer architecture appears after the multi-head self-attention layer. Although it is based on the feed-forward module, it has a nonlinear activation function called Swish activation [30] between the two linear transitions. A dropout is also added to normalize the network. The module follows the feed-forward module in the transformer architecture and uses the pre-norm residual unit as shown in Figure 12.

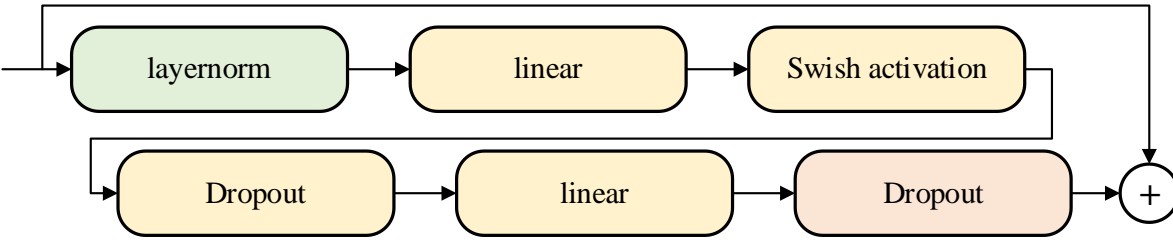

**Figure 12.** Feed-forward module.

The conformer training data set for this paper is the entire training set of LibriSpeech, which is approximately 1000 h long. The audio is extracted one frame every 25 ms, with each frameshift at 10 ms. The 80-D audio spectrum is obtained by 80-channel filter banks and the speech signal is enhanced by using Spec Augment [31,32].

In this paper, the conformer used character error Rate (CER) and word error rate (WER) [33] for testing. We focused mainly on the WER, in which the lower the WER, the better the training of the model. The Equation for the calculation of the WER is shown in Equation (10):

$$WER = \frac{(S + D + I)}{N} \times 100\% = \frac{S + D + I}{S + D + C} \times 100\% \tag{10}$$

where S denotes the number of substituted words, D is number of deleted words, I is the number of inserted words, N is the total words in original text, and C is the correct number of words.

### 2.3.4. Model Testing Result

In this section, the training results of the two models are presented: (1) the SDR with speaker separation model and (2) the WER with speech recognition mode.

(1) Speaker Separation Model Testing

There are different training sets for evaluating the performance of the speaker separation model. Table 3 shows the results of the different training sets (i.e., train-clean-100, train-clean-360, train-clean-100, and 8 train-clean-360).

**Table 3.** Evaluation of speaker separation model.

| Training Set | SDR | Frequency (Step) | Batch Size |
|---|---|---|---|
| Train-100 | 3.2 dB | 65 k | 8 |
| Train-360 | 5.4 dB | 65 k | 8 |
| Train-100 and train-360 | 7.5 dB | 30 k | 8 |
| Train-100 and train-360 | 8.6 dB | 530 k | 4 |

In the first experiment, the performance results of the train-clean-100 dataset with 251 speakers were not good with an SDR of only 3.2 dB. Further, the actual output audio was much different from the clean audio. Moreover, the output volume was about 80% less than the input volume and the specific speaker audio was not separated very successfully, and is similar to turning down the volume of the mixed audio. The training effect of the train-clean-360 with 921 speakers was slightly improved; the volume was still cut, but the magnitude was reduced and the separation effect was slightly present. As can be seen in Figure 13, the SDR was raised to 5.4 dB, and the volume of speaker A was greater than the volume of speaker B. Finally, the two training sets were summed up and the number of speakers reached about 1200. The training was conducted twice. The first time we set the batch size to 8, the SDR stopped at about 7.5 dB at the frequency step of 300 k; so, we lowered the batch size to 4, and the SDR continued to set at 300 k and approached a better SDR of 8.6 dB at 530 k. As shown in Figure 14, the separated clean audio was precisely matched to the actual output. The proposed effectiveness of the separated model was to reduce the influence of the outside environment. This experiment also illustrated that the more speakers are included in the training set, the better the performance of the separated trained model. This characteristic of the speaker separation model also supports the next ASR model to obtain the adaptive reality to fit the right voice command in a real environment.

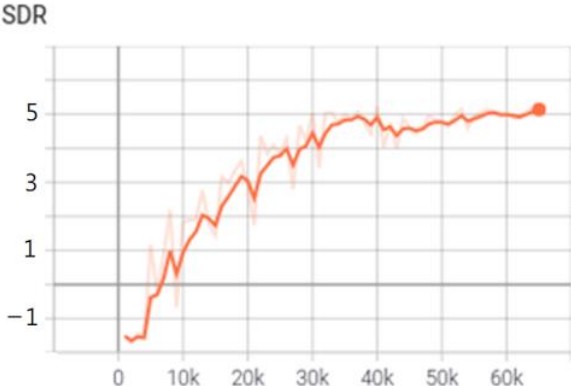

**Figure 13.** The related SDR value with the amount dataset of 921 people.

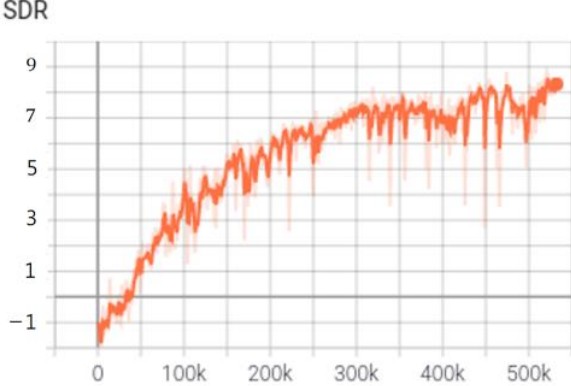

**Figure 14.** The related SDR value with the amount dataset of 1200 people.

(2)    Evaluations for Automatic Speech Recognition Model

The model discussed earlier was added to the ASR system. The training data were divided into several different sets to evaluate the goodness of WER. All the training cycles were set as epoch = 100 and batch size = 4. The audio was added to the noise to obtain the adaption of the speaker separation. The test set with a different amount of training dataset was mainly applied to evaluate the ASR model (see Table 4). When the total training cycle for train-100 audio data was finished, it obtained a not-so-good WER of 14%. In another experiment, the total training cycle was gradually added to the other training data set (i.e., train-360 and train-500) to approach the best WER, which was from 14% to 5.3%. The smallest unit in the English language is a single word, so most of the Speech Recognition model uses a WER index to detect the performance of the trained result. In this paper's real speech recognition experiment, a total of 129 different English voice patterns were continuously collected from different individuals and the pre-trained speech recognition model built in this study was applied to finally approach the correct rate of 89.3% in the real environment.

**Table 4.** Evaluations of automatic speech recognition model.

| Training Set | WER Test Result |
| :---: | :--- |
| Train-100 | 14.1% |
| Train-360 | 11.5% |
| Train-100 and train-360 | 10.3% |
| Train-100 and train-360, and train-500 | 5.3% |

Zhang et al. used the semi-supervised learning method with the large-scale YouTube-based data to pre-train the conformer model based on the original LibriSpeech database and the upstream/downstream to self-train the model size, so it will match the multiple public datasets. Experiential results reached the best WER of 7.7% as it approached multiple tasks by voice command [34]. Several more studies have registered and trained the deep neural network HMM to advance ASR in a real environment. A pretraining ASR system has been built with a novel learning stratagem and with a structure of the artificial neural network (ANN) model based on a large, but limited playback loudspeaker sub databases. It was aimed to outperform the famous and appropriated Google API, IBM API, and Bing API in terms of the WER [35] after completing an environment-based training cycle with the specific questionnaires. In another voice command with robot application study [36], researchers provided a cloud-based NLP platform and ROS-based mobile robot system for participants interacting with the robot. The lowest WER it obtained was 5.8% and the highest was 83.3% with the correct rate of voice entity detections in an office, without an ambient noise environment. The WER increased to 21.9% and the correct rate of entity detections was decreased to 61.7% with respect to the effect of noise. In the research paper [37], receiving and recognizing voice commands were presented using a LabVIEW machine learning model for a person to communicate with the robot. Its practical tests, voice command reached an average precision of 86% for "go forward and back", and 78% correctness for "left and right" to control the robot's actions. Its total average precision reached 82%, while people used voice to control the mobile robot forward and backward. A multi-stream HMM decision fusion [38] was also used to improve the voice recognition rate of the acoustic model to 88.5% in the close-talking type of ASR experiments. The other identification of voice correctness was approximately 72.4% when the throat microphone was used to test the same voice command.

Table 5 compares the evaluations of WER and speech recognition accuracy rates in many different methods. This data analysis illustrates that the proposed deep-learning type voice machine achieved the best results with the lowest WER of speech recognition testing and the highest accuracy in control command for the mobile robot. A generalized speech training database can result in more complete word comprehension, but requires more intensive training methods and more training time. Single-voice training is easier because it is based on a specific set of required commands. This research paper suggests that the performance of the general-purpose speech training database is the best result compared to other research papers.

**Table 5.** Comparisons for WER and accuracy of speech recognition in different methods.

| Methods | WER% | Accuracy of Speech Recognition | Fitting Type |
|---|---|---|---|
| Zhang et al. [34] | 7.7% | X | Universal word type |
| Novoa et al. [35] | 11.62% | X | Universal word type |
| Google API [35] | 15.79% | X | Universal word type |
| IBM API [35] | 40.74% | X | Universal word type |
| Stuede et al. [36] | 5.8% | 83.3% | Single word type |
| Pleshkova et al. [37] | X | 82% | Single word type |
| Heracleous et al. [38] | X | 88.5% | Single word type |
| The proposed methods | 5.3% | 89.3% | Universal word type |

### 3. Environment Map Generation through VSLAM

A real-time appearance-based mapping (RTAB-Map) algorithm based on the concept of SLAM to generate the map of the environment is proposed in this study. The RGB-D cameras concurrently obtain the image and distance information. The RGB-D maps do not consume much computational time to calculate the distance, and approach better accuracy of position than the binocular or monocular camera.

The proposed RTAB-Map SLAM algorithm is divided into three parts: (1) front terminal part, (2) rear terminal part, and (3) loop closure detection. The objective of visual odometry extracts the feature and information from the odometer's captured image, and plots the local map. The rear part is used to optimize the original maps and robot posture estimation and other processes. Figure 15 shows the Flow Chart of the RTAB-based SLAM.

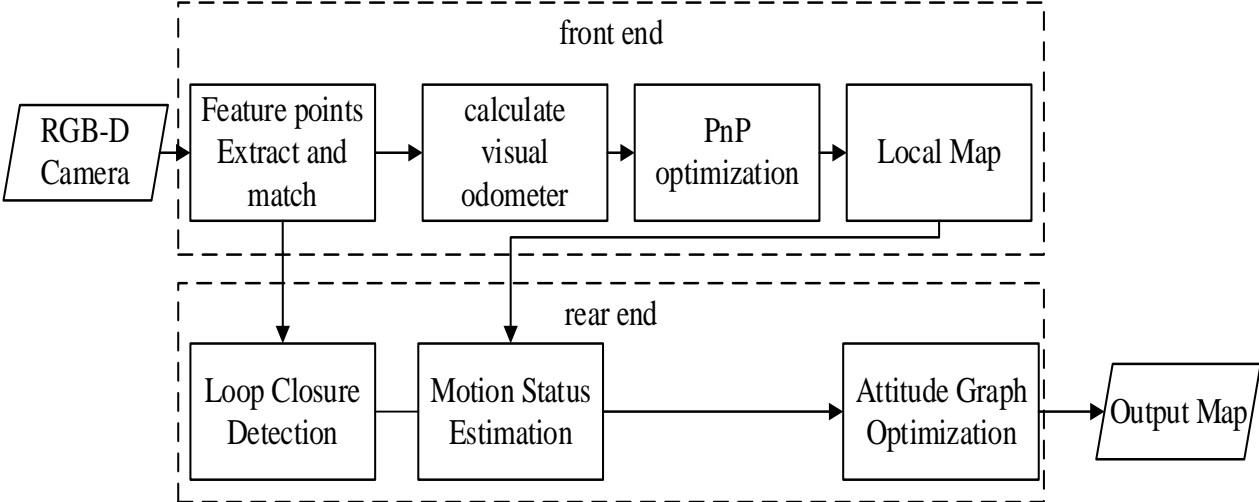

**Figure 15.** Flow chart of the RTAB-based SLAM.

In the rear part, the speeded-up robust features (SURF) is used to extract and match the feature point purpose [33]. A Hessian matrix is applied to detect feature points; image integration is also proposed to speed up the calculation. The URF structure is a multiple-layer filter containing different filters in the same layer. Further, same-sized filters are contained in the different layers of URF. Therefore, the Vague coefficient of filter is gradually increased to reduce the captured process and improve the handing speed. SURF confirms the matching degree based on the Euclidean distance between two feature points, where the shorter distance presents the better matching degree.

To calculate the visual odometry, the reference frame is obtained from the previous time and the current frame is defined in the now time. The reference frame is selected as the benchmark of the coordinate system, where it is matched with the current frame. The random sample consensus-perspective-n-points (RANSAC-PnP) [39] is taken to calculate the coordinate position of the current RGB-D sensor. The previous cycle is then repeatedly performed while the whole movement track of the RGB-D sensor is achieved. This path is considered the track of the mobile robot.

The local map is selected as the matched feature point of the object of interest in the environment, and the real position and posture of the object are calculated through the matching procedure between the current frame and the feature point. The advantage of the local map method is that it can be modified based on the constructed information of each frame even when some of the resulting frames are wrong.

In the rear terminal part, the SLAM procedure meets the error noise problem, which is gradually accumulated when operating in the front part. The error is a big amount because it is accumulated over a long period. Therefore, it appears that there is a huge gap between the constructed and the real maps; thus, collecting all the possible map information is necessary to complete the whole optimization. The optimization is added to the load tags from the previous motion trajectory, which is obtained in the front part; the prediction of the motion state is then approached to the desired target. After that, the second rank in optimization is achieved to obtain the better pose graph, which is needed to optimize the global map after updating the motion state. Because the object is continuously detected by the RGB-D sensor, the same object is always constructed, which causes some errors in the map. This research extracted the primary feature point to match the object and then, the position value was attained through the motion relationship. Through this, we were able to efficiently construct the dynamic motion of the mobile robot while generating and keeping the trajectory of the primary feature point.

For the loop closure detection function, this study proposed the RTAB-map algorithm [15] to finish the real-time localization function of the robot. Figure 16 shows the flowchart of the RTAB-map algorithm.

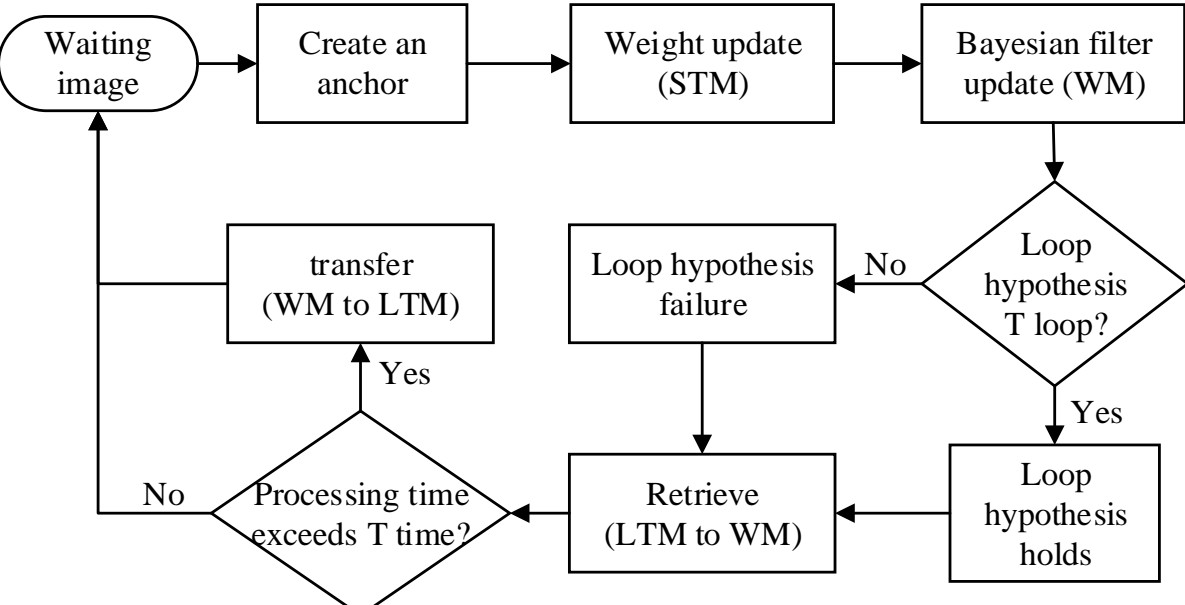

**Figure 16.** Flow chart of the loop closure detection by RTAB-map.

In creating the robot's localization point, the bag-of-words without the process of training the specific environment is used to obtain favorable localization data. The new localization weight is updated in the next stage. The current localization point makes a similarity comparison with the last localization point of the short-term memory (STM). The current localization is replaced by the last localization point of STM if the similarity is over the defined threshold.

The Bayesian filter is used to refresh the working memory (WM). It evaluates the probability of loop closure between the current localization point and the localization point of WM. The closure loop with the highest p is established, with the probability p less than the defined threshold. The connection between the new and old localization points is ensured, and the new weight is refreshed by adding the original weight to the old one.

The localization points with the lowest weight and the longest holding time are transferred into long-term memory (LTM) [40] with the graph processing time set at an interval time T. Thus, the inverse distance weighting method is used to reorganize the weight to greatly improve the efficiency of the closure loop detection. Figure 17 illustrates the simulation of the feature-matching result.

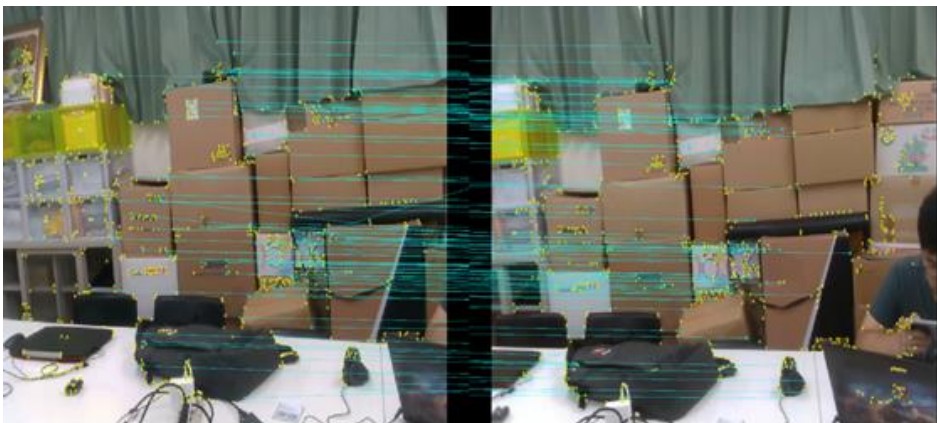

**Figure 17.** Simulation for feature-matching result.

## 4. Voice Interactive Robot Control in Real-Robot Experiment

In the last real-world experiment, the speaker separation and ASR are combined to accomplish the voice interactive-based mobile robot applications. The speaker separation can percolate the voice into a specific signal and the ASR analyses the signal to transfer it as a control command. A detailed mobile robot with dual-arm architecture is built in our laboratory [41,42], which completes the object recognition and placement at the set position for different specific service tasks in a real environment. Detailed demonstration of this paper in a real environment is illustrated in the next paragraph.

Figure 18 shows the service tasks through a real robot that can correctly pick-up and deliver three types of objects based on the human voice control command. A mobile robot stands in front of the service desk, and is commanded by a human voice to pick up and deliver three types of objects: an alcohol bottle, a box, and a toilet paper. Figure 18a shows the mobile robot receives the control command "give me the alcohol", and Figure 18b illustrates the mobile robot makes the next action decision by recognizing the control command, which is to pick up the alcohol bottle in Figure 18c. Figure 18d shows the robot delivers the bottle to the service request in Figure 18e. The sequential actions presented in Figure 18f–j exhibit that the service robot successfully picks up and delivers the box to the correct position after recognizing the voice command "give me the box". Finally, Figure 18k–o show the mobile robot completing the service task as a response to the voice command "give me the toilet paper". After giving the user toilet paper, Figure 18p shows that the mobile robot moves back to the original location and stands by. The demonstration for the service mobile robot systems in a real environment are fully illustrated in detail in the video presentation, indicating a successful human-interactive robot application in an external noise environment [43]. In addition to the service actions shown above, the dual-arm mobile robot can perform the other different behavioral actions, such as cup recognition and gripping, according to the specific actions that need to be controlled with human voice after completing the manipulator setting and training for these tasks.

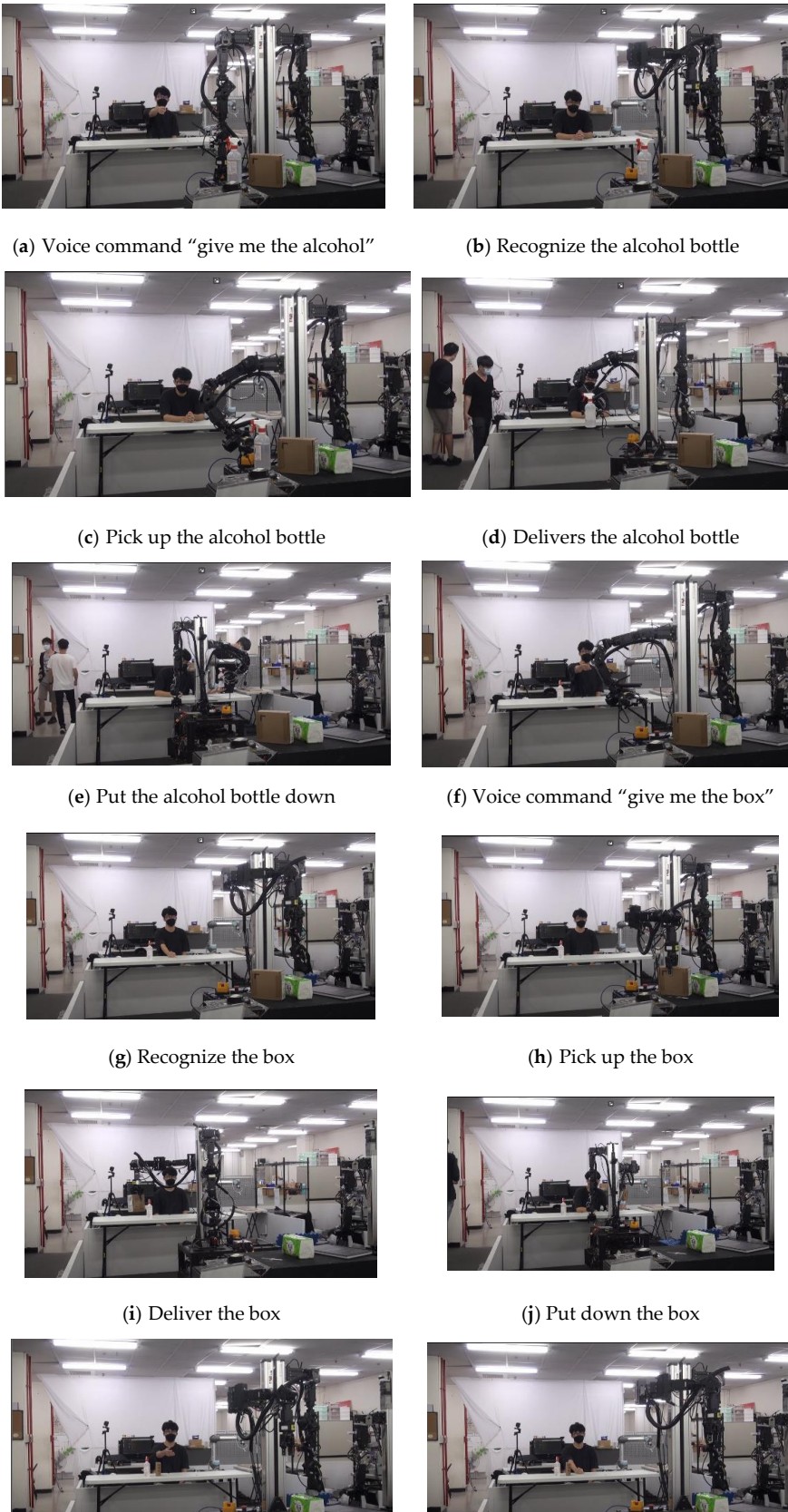

(**a**) Voice command "give me the alcohol" (**b**) Recognize the alcohol bottle

(**c**) Pick up the alcohol bottle (**d**) Delivers the alcohol bottle

(**e**) Put the alcohol bottle down (**f**) Voice command "give me the box"

(**g**) Recognize the box (**h**) Pick up the box

(**i**) Deliver the box (**j**) Put down the box

(**k**) Voice command "the toilet paper" (**l**) Recognize the toilet paper

**Figure 18.** *Cont.*

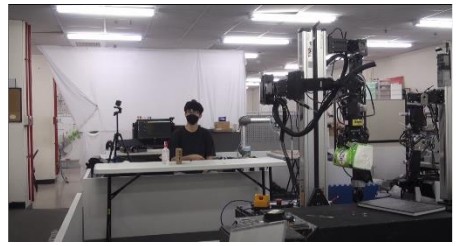
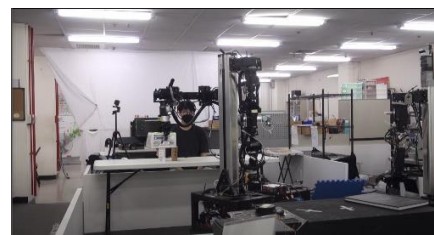

(**m**) Pick up the toilet paper    (**n**) Deliver the toilet paper

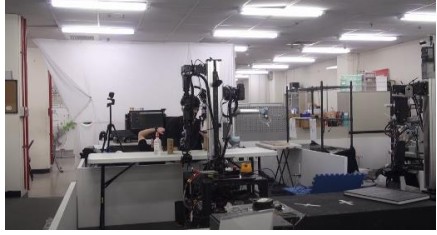
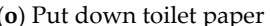
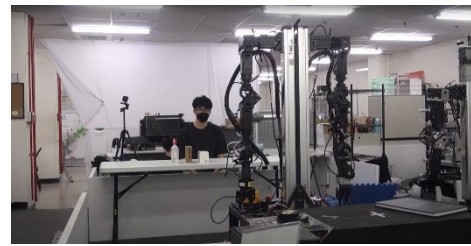

(**o**) Put down toilet paper    (**p**) Moves back to the original location

**Figure 18.** Sequential actions of a real mobile robot for different service tasks.

## 5. Conclusions

This paper combined the speaker separation system and ASR to analyze the practical semantic word of human speech. The speaker separation system included a fast speaker encoder, which requires a lesser time to convert the speech spectrum and remove the ambient noise using a trained voice filter. Thus, a clear and correct control instruction can run through a noisy environment to automatically finish speech recognition. The conformer module used a block CNN-type encoder to form a transformer framework, which takes the advantage of the multi-head self-attention machine to note the affirmatory word, efficiently separate the concise word, and clearly acquire the correct command to handle the required actions for the mobile robot.

A real-time vision-based appearance-base mapping (RTAB-map) algorithm, which utilized the RGB-D device, was employed in this study to concurrently generate the environmental map and fit the matching boundary of the object in the working space. The RTAB-map with a visual odometer was employed to predict the motion of the mobile robot to appropriately guide it into the unloading zone and service area to complete the navigation process.

Finally, a real mobile robot with dual arm was used to test the study's model. It was placed in front of a table and was asked to complete a series of human voice-activated commands. The results showed that the robot was able to correctly recognize the voice command and efficiently perform the service tasks.

**Author Contributions:** Conceptualization, S.-A.L. and H.-M.F.; methodology, S.-A.L., Y.-Y.L. and H.-M.F.; software, Y.-Y.L. and Y.-C.C.; validation, S.-A.L., Y.-Y.L. and Y.-C.C.; formal analysis, S.-A.L. and Y.-Y.L.; investigation, Y.-C.W. and P.-K.S.; resources, Y.-C.W. and P.-K.S.; data curation, S.-A.L. and H.-M.F.; writing—original draft preparation, Y.-Y.L. and Y.-C.C.; writing—review and editing, S.-A.L. and H.-M.F.; funding acquisition, S.-A.L. and H.-M.F. All authors have read and agreed to the published version of the manuscript.

**Funding:** This paper was partly supported by the Ministry of Science and Technology of the Republic of China with contract number: MOST-110-2221-E-032-039 and MOST-110-2221-E-507-009.

**Institutional Review Board Statement:** Not applicable.

**Informed Consent Statement:** Not applicable.

**Data Availability Statement:** Not applicable.

**Conflicts of Interest:** The authors declare no conflict of interest.

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
