# Peer review of "Voice Interaction Recognition Design in Real-Life Scenario Mobile Robot Applications"

_applsci, doi:10.3390/app13053359_

Round 1

Reviewer 1 Report

The paper presents the speech recognition system for a mobile robot. The authors give are survey of speech recognition methods, describe their speaker separation and speech recognition systems, and show illustrates of experiments of voice robot control in real-life scenario.  

There is no description of the robot in the paper. It is not clear what it is intended for, what commands it can recognize. The paper shows the examples of recognition of three commands: “give me the alcohol”, “give me the box”, “give me the toilet paper”. Is that all the robot can do?

In the paper, the detail description of speaker separation and speech recognition system is given. However, the applied methods are quite standard. What is the novelty of used methods? What is the specific in using these methods exactly for voice robot control? The speech recognition system was trained on LibriSpeech corpus, which is very popular corpus for training the speech recognition systems, however the phrases in this corpus are not from robot control domain. Thereby, the created speech recognition system has not specifics for mobile robot control domain. And it is not clear, why the authors focus on the control of mobile robot.

It is worth presenting a survey of mobile robots with voice control and compare your solution with existing ones.

Author Response

To Reviewer #1

We are also extremely grateful to the reviewers’ comments on our revised manuscript and carefully considered every comment, and made cautious revision accordingly. Based on their suggestions, we have answered all these questions in detail one by one. If you have any other questions about this paper, I would quite appreciate it if you could let me know them in the earliest possible time.

Most sincerely,

Hsuan-Ming Feng

Our response is explained in the followings:

1. There is no description of the robot in the paper. It is not clear what it is intended for, what commands it can recognize. The paper shows the examples of recognition of three commands: “give me the alcohol”, “give me the box”, “give me the toilet paper”. Is that all the robot can do?

ANS: We thank the reviewers for their suggestions and comments. The structure of the robot in this paper is based on the Dual-Arm mobile robot, which is support by the previous laboratory equipment [42-43]. This research uses this as a basis and combines with the voice recognition system to provide human-robot interaction for completing the service control of the mobile robot in a real environment. In addition to the service actions shown above, the Dual-Arm mobile robot can also perform different behavioral actions, such as cup identification and grasping, depending on the specific training actions that need to be controlled. Please check pages 530-533 of the revised paper.

2. In the paper, the detail description of speaker separation and speech recognition system is given. However, the applied methods are quite standard. What is the novelty of used methods? What is the specific in using these methods exactly for voice robot control?

The speech recognition system was trained on LibriSpeech corpus, which is very popular corpus for training the speech recognition systems, however the phrases in this corpus are not from robot control domain. Thereby, the created speech recognition system has not specifics for mobile robot control domain.

And it is not clear, why the authors focus on the control of mobile robot.

It is worth presenting a survey of mobile robots with voice control and compare your solution with existing ones.

ANS:

(1)The novelty of this study is to obtain a generic speech recognition model by training phrases described in the generic LibriSpeech corpus, and to use the pre-trained Conformer model structure to make human-robot interaction resistant to interference from the external environment by designing speech separation filters. At the same time, the accuracy of the real environment is used to achieve better results than other mobile robot speech control systems.

 (2) Table 5 compares the evaluations of WER and speech recognition accuracy rates in many different methods [36-39]. In particular, the accuracy of our proposed robot speech control is better than these literatures [37-39].

(3) We added more descriptions on pages 404-433.

Reviewer 2 Report

The paper is well structured. The introduction provides a comprehensive explanation of the entire work and contextualizes it correctly. I have missed adding a description with the structure of the paper at the end of the introduction.

The number of references is adequate and all the methods are correctly introduced, cited or explained throughout the paper.

The Latex format does not comply with the standard since the line numbers from section 2 do not appear.

The entire document is grammatically well written and the English is correct, however, I have found the following aspects that should be improved:

Line 37 Dynamic Time Warp (DTC) --> DTW

Line 137 The ROS --> Only ROS without The

Change occurrences of Formula --> Equation

Add a Link (hyperlink) to each equation, figure and table.

I miss a brief introduction of the robot explaining its general characteristics.

Through the figures and the video of the tests, it is interpreted that the objects are placed in known points. That is, it does not determine its position in space by artificial perception. Therefore, it is also interpreted that the destination position of the robot (goal) is established based on the object it wants to pick up.

Although it is not the purpose of this paper, it is advisable to make it clear.

Is the recognition semantic or does it search the interpreted text for reserved words (alcohol, toilet paper, box)? For example, if you say to the robot "bring me the spray" it would understand it or wait for the reserved word (alcohol).

In general I think it is a good work and can be published after these slight changes.

Author Response

To Reviewer 2

We are also extremely grateful to the reviewers’ comments on our revised manuscript and carefully considered every comment, and made cautious revision accordingly. Based on their suggestions, we have answered all these questions in detail one by one. If you have any other questions about this paper, I would quite appreciate it if you could let me know them in the earliest possible time.

Most sincerely,

Hsuan-Ming Feng

Our response is explained in the followings:

1. The paper is well structured. The introduction provides a comprehensive explanation of the entire work and contextualizes it correctly. I have missed adding a description with the structure of the paper at the end of the introduction.

ANS: We added this description on pages 158-161 of the revised paper. Please check.

2.The Latex format does not comply with the standard since the line numbers from section 2 do not appear.

ANS: We have filled in the required line numbers.

3. The entire document is grammatically well written and the English is correct, however, I have found the following aspects that should be improved:

Line 37 Dynamic Time Warp (DTC) --> DTW

Line 137 The ROS --> Only ROS without The

Change occurrences of Formula --> Equation

Add a Link (hyperlink) to each equation, figure and table.

ANS: English corrections and line number display have been completed. Added corrections such as links (hyperlinks) to each equation, chart and table.

Reviewer 3 Report

This paper describes the design of a voice interactive robot system that can execute service tasks in real-life scenarios. Special attention is paid to the voice filter, ASR and environment map generation modules. This is a significant R&D effort and deserves to be published after addressing some issues. The main drawback is the experiment section which is really a demo. It does not correspond to results obtained with a database recorded in the real HRI scenario. Of course, this is out of the scope of the peer review window. Instead, the authors can always justify better each module mentioned above to achieve the final target. Also, the experiment section could be renamed as a “demo in real environment”. My specific comments are the following ones:

1-The authors should clarify the relationship between AI and HMM.

2-The reference is complete. However, the importance of voice based HRI can be emphasized with [1].

3-The description of the traditional ASR technology needs a significant improvement. Acoustic modelling is not carried out only with HMM (or GMM) now days. Actually, replacing GMM with DNN is a clear example. Also, the traditional decoding process (usually with the Viterbi algorithm) is the one that delivers the most likely sequence of words by employing the acoustic and language modelling.

4-The English writing can be improved. The authors should consider to ask an English native speaker or a professional service to proofread the manuscript.

5-In my opinion, the authors dedicate too much room to explain concepts that should be known by the target readers. For instance, Gaussian or Normal p.d.f. and  SDR.

6-I am wondering if Table 1 is really needed since it does not show results reported in the manuscript.

7-Table 5 is very interesting as a reference to compare different initiatives toward robust voice based HRI. However, the authors need to observe in the paper that the straight comparison is not always possible due to the fact that the training and testing conditions are different. This is why speech databases are delivered to evaluate speech technologies, including ASR.

8-The evaluation of the ASR module is not clear: “The test set with a different data number was mainly applied to evaluate the ASR model (see Table 4).” What do you mean with “different data number”? I could not find in Table 4.

8-Figure 17 is also very interesting. However, the caption should say what is going on each subplot. Also, subplot “j” is empty. Could not you use an even number of subplots?

Reference

[1] Wuth, J., Correa, P., Núñez, T., Saavedra, M. Yoma, N.B. “The Role of Speech Technology in User Perception and Context Acquisition in HRI.” International Journal of Social Robotics. April 2020.

Author Response

To Reviewer 3

We are also extremely grateful to the reviewers’ comments on our revised manuscript and carefully considered every comment, and made cautious revision accordingly. Based on their suggestions, we have answered all these questions in detail one by one. If you have any other questions about this paper, I would quite appreciate it if you could let me know them in the earliest possible time.

Most sincerely,

Hsuan-Ming Feng

For review’s response list, please check follows:

1. I miss a brief introduction of the robot explaining its general characteristics. Through the figures and the video of the tests, it is interpreted that the objects are placed in known points. That is, it does not determine its position in space by artificial perception. Therefore, it is also interpreted that the destination position of the robot (goal) is established based on the object it wants to pick up. Although it is not the purpose of this paper, it is advisable to make it clear. Is the recognition semantic or does it search the interpreted text for reserved words (alcohol, toilet paper, box)? For example, if you say to the robot "bring me the spray" it would understand it or wait for the reserved word (alcohol).

ANS:

We thank the reviewers for their suggestions and comments.

The structure of the robot in this thesis is based on the Dual-Arm mobile robot designed by the previous lab and combined with the speech recognition system in this thesis to provide human-computer interaction to complete the service control of the mobile robot in a real environment [42-43], which are described in the revised paper on pages 510-514. The tasks of the different services are also described in additional detail on pages 529 to 532 of this revised paper. Please check!

2. The authors should clarify the relationship between AI and HMM.

ANS: We have revisited and corrected it on pages 38-42.

3. The reference is complete. However, the importance of voice based HRI can be emphasized with [1]

ANS: We have revisited and added this reference in the revised paper[2].

4. The description of the traditional ASR technology needs a significant improvement. Acoustic modelling is not carried out only with HMM (or GMM) now days. Actually, replacing GMM with DNN is a clear example. Also, the traditional decoding process (usually with the Viterbi algorithm) is the one that delivers the most likely sequence of words by employing the acoustic and language modelling.

ANS: We have modified some of the elucidations in this paragraph from pages 89-112. 

5. The English writing can be improved. The authors should consider to ask an English native speaker or a professional service to proofread the manuscript.

ANS: We have asked the relevant native language professionals to edit.

6. In my opinion, the authors dedicate too much room to explain concepts that should be known by the target readers. For instance, Gaussian or Normal p.d.f. and SDR.

ANS: Because the readership of this paper contains a number of mobile robotics-related professionals who may need a clearer understanding of the topic, the relevant descriptions are reserved for the time being.

7. I am wondering if Table 1 is really needed since it does not show results reported in the manuscript.

ANS: Table 1 will be retained because it is used in this paper to demonstrate the design of diluted convolutional layers to more efficiently extract low-level acoustic features.

8. Table 5 is very interesting as a reference to compare different initiatives toward robust voice based HRI. However, the authors need to observe in the paper that the straight comparison is not always possible due to the fact that the training and testing conditions are different. This is why speech databases are delivered to evaluate speech technologies, including ASR.

ANS: We have marked the characteristics of the phonetic database and distinguished between individual specific words and general-purpose training words, which are generally more complex and more difficult to be trained, and have added a description on pages 429-433 of the paper.

9. The evaluation of the ASR module is not clear: “The test set with a different data number was mainly applied to evaluate the ASR model (see Table 4).” What do you mean with “different data number”? I could not find in Table 4.

ANS: It is modified into “different amount of training dataset”

10. Figure 17 is also very interesting. However, the caption should say what is going on each subplot. Also, subplot “j” is empty. Could not you use an even number of subplots?

ANS: We describe on page 515-523 what is going on in each subgraph and add a subgraph where the robot returns to the home position and waits for the command.  

Round 2

Reviewer 1 Report

The authors have incorporated all the necessary changes and corrected all reviewer's comments.  The revised version of the paper is now more comprehensive and informative.

I think the paper can be accepted for publication in Applied Sciences.